# Heterogeneous Impact of Water Warming on Exotic and Native Submerged and Emergent Plants in Outdoor Mesocosms

**DOI:** 10.3390/plants10071324

**Published:** 2021-06-29

**Authors:** Morgane B. Gillard, Jean-Pierre Caudal, Carole Deleu, Gabrielle Thiébaut

**Affiliations:** 1ECOBIO, UMR 6553 CNRS, Université de Rennes 1, 35042 Rennes, France; jean-pierre.caudal@univ-rennes1.fr (J.-P.C.); gabrielle.thiebaut@univ-rennes1.fr (G.T.); 2IGEPP, UMR 1349 INRAE, Institut Agro Agrocampus Ouest, Université de Rennes 1, 35653 Le Rheu, France; carole.deleu@univ-rennes1.fr

**Keywords:** climate change, species biogeographic origin, biological traits, phenology, macrophyte growth form

## Abstract

Some aquatic plants present high biomass production with serious consequences on ecosystem functioning. Such mass development can be favored by environmental factors. Temperature increases are expected to modify individual species responses that could shape future communities. We explored the impact of rising water temperature on the growth, phenology, and metabolism of six macrophytes belonging to two biogeographic origins (exotic, native) and two growth forms (submerged, emergent). From June to October, they were exposed to ambient temperatures and a 3 °C warming in outdoor mesocosms. Percent cover and canopy height were favored by warmer water for the exotic emergent *Ludwigia hexapetala*. Warming did not modify total final biomass for any of the species but led to a decrease in total soluble sugars for all, possibly indicating changes in carbon allocation. Three emergent species presented lower flavonol and anthocyanin contents under increased temperatures, suggesting lower investment in defense mechanisms and mitigation of the stress generated by autumn temperatures. Finally, the 3 °C warming extended and shortened flowering period for *L. hexapetala* and *Myosotis scorpioides*, respectively. The changes generated by increased temperature in outdoor conditions were heterogeneous and varied depending on species but not on species biogeographic origin or growth form. Results suggest that climate warming could favor the invasiveness of *L. hexapetala* and impact the structure and composition of aquatic plants communities.

## 1. Introduction

In freshwater aquatic systems, high biomass development of macrophytes (aquatic plants) forms dense mats that can generate waterways obstruction, changes in water quality, or impact biodiversity [1,2]. Macrophyte species can only grow into nuisance stands if hydromorphological conditions are suitable, if the supply of nutrients and light is sufficiently high to enable massive growth and if simultaneous losses by, e.g., herbivory are low. Furthermore, mass development also depends on plants’ performance under local climatic conditions [3].

Temperature is one of the main factors influencing biochemical reactions. Therefore, slight changes in temperature can have substantial impacts on physiological processes such as photosynthesis [4], respiration [5], or nutrition status [6]. Global surface temperatures are predicted to rise by up to 4 °C by 2100 [7], and such increased temperatures are expected to modify species fitness, measurable through morphological and physiological traits [8]. For example, photosynthetic pigment contents are often impacted by changes in environmental conditions [9,10]; they often decrease under heat stress [11], and are good proxies for the photosynthetic capacity [12]. Since there is a close interrelationship between photosynthetic activity and growth, reduced photosynthetic pigment contents induced by warming greatly impact growth and biomass production. Defense compounds such as flavonols or anthocyanins are also regulated by environmental factors and are therefore good candidates to evaluate plant response to temperature [13,14], along with other biochemical parameters such as allocation of soluble sugars or starch that reflect impacts on plant carbon metabolism [15,16]. The changes generated by temperature increase on macrophyte species can differ depending on seasons [17,18] and studying warming impacts over a growing season allows the exploration of phenological changes. To our knowledge, relatively few studies have been conducted on the impact of climate warming on macrophytes phenology. Zhang et al. (2016) [19] showed that temperature increase significantly advances shoot emergence of *Potamogeton crispus*, while Rooney and Kalff (2000) [20] recorded a more precocious growth in warmer years for communities of submerged macrophytes. Calero and Rodrigo (2019) [21] demonstrated that temperature greatly contributes to flowering variability, with varying impacts depending on species within a submerged macrophyte community. The modifications provoked by warmer environmental conditions can be favorable or detrimental depending on aquatic plant species [22] and on their exotic or native status [23,24]. Indeed, invasive populations often present high adaptive plasticity compared to natives [25], providing them with the ability to invade climatic niches quite different than those of their area of origin [26,27]. Nonetheless, plant invaders do not systematically benefit from climate warming more than natives [28,29]. Thus, exploring responses to increasing temperatures of a variety of species is essential to predict potential changes in community structure and composition.

For macrophytes, warming may have different impacts depending on growth form [24,30], submerged growth form being completely underwater, while emergent species are exposed to both air and water. Therefore, due to the inherent thermal inertia of water, at daily timescale submerged species are less exposed to temperature variations than emergent species, and so are likely less susceptible to heatwaves for example. However, at a seasonal or yearly timescale, they could be more sensitive to warmer water temperatures. Growth form has been shown to influence temperature within macrophyte beds [31,32]; however, the study of comparative responses of different growth forms to an overall temperature increase is scarce.

We aimed to investigate morphological, physiological and phenological responses of macrophytes belonging to two biogeographic origins (exotic, native) and to two growth forms (submerged, emergent) when exposed to increased water temperatures for several months. We exposed six species to water at ambient temperature and to water warmed 3 °C above ambient temperature water, for about five months in outdoor mesocosms. We hypothesized that: (1) warmer temperature will impact both growth, metabolism and phenology; (2) exotic species will not benefit more from the warming than native species; and (3) the impacts of water temperature increase will be greater on submerged species than on emergent ones.

## 2. Results

### 2.1. Morphological Traits

Over the 7 weeks of the acclimation period, the number of individuals per mesocosm decreased, with the loss of about 0 to 40% of individuals. *Egeria densa* seemed to be producing fragments, and this tendency could have been favored by the repetitive identification and counting of rooted individuals, despite careful manipulation. Therefore, for this species, the monitoring of the number of individuals was rapidly stopped and the number of individuals was considered stable. For other species, the number of individuals slightly declined during the experiment, with a decrease of one individual per mesocosm on average (Figure 1, Table 1). There was no impact of temperature on the number of individuals.

The percent cover significantly increased over time for all six species, with an interaction between species and temperature (Figure 2a, Table 2). There was no effect of treatment T3 on the percent cover of *Myriophyllum aquaticum*, *Mentha aquatica*, *Myosotis scorpiodes* and *P. crispus* (Figure 2a). However, after about 120 days of exposure to T3, the percent cover of *Ludwigia hexapetala* became significantly 44% greater in mesocosms exposed to warmer water temperature compared to those exposed to T0. At the end of the experiment, after 140 days of exposure to warmer conditions, the percent cover of *L. hexapetala* was 52% greater than in the ambient temperature condition.

On the contrary, after 120 days of exposure to T0 and T3, the percent cover of *E. densa* was significantly 1.9-fold greater in mesocosms exposed to T0 (Figure 2a), a difference that was maintained after 140 days of exposure to the differential treatments. Fixed effects explained 42% of the variability in species percent cover, mainly driven by time and species, while the random effect (mesocosm) explained another 19% of the variability.

Maximum canopy height of the four emergent species varied depending on species, time and temperature (Table 2). For *M. aquaticum* and *M. scorpioides*, the maximum canopy height was stable over time, independent of temperature (Figure 2b). For *M. aquatica* and *L. hexapetala*, the maximum canopy height increased continuously over the experiment, although there was no impact of water temperature on *M. aquatica*. However, after about 90 days of exposure to T3, the maximum canopy height of *L. hexapetala* was greater in this condition than when exposed to T0 (Figure 2b), and was 39% greater at the end of the experiment. Fixed effects explained 57% of the variability in maximum canopy height, largely driven by species, while the random effect explained 5% of the variability. After 20 weeks of exposure to the two temperature treatments, the total dry biomass was not different depending on temperature conditions (Figure 3, Appendix A).

### 2.2. Content Evaluation of Some Metabolic Compounds

There was no impact of temperature on chlorophyll content and NBI (Figure 4a, Appendix A). Anthocyanins content of *M. aquatica* decreased by 53% at T3. Flavonols content was significantly lower at T3 for the three species, with a decrease of 2% for *L. hexapetala*, 21% for *M. scorpioides*, and 12% for *M. aquatica* (Figure 4a).

Overall, total soluble sugars were 23% lower when plants had been exposed to a 3 °C warming, independent of species (Figure 4b, Appendix A). The overall amount of total soluble sugars in *M. scorpioides* was 1650 µmol-g^−1^ DW, a value 2.6-fold to 7.8-fold greater than that of the five other species. *M. aquatica* presented the second-highest amount in total soluble sugars, ≈630 µmol-g^−1^ DW, independent of temperature. The four other species all had mean total soluble sugars amounts between 200 and 300 µmol-g^−1^ DW, with *L. hexapetala* presenting the highest values, *M. aquaticum* and *P. crispus* presenting the lowest values and *E. densa* showing intermediate values.

### 2.3. Flowering Phenology

Four species produced flowers during the experiment: *E. densa*, *L. hexapetala*, *M. aquatica* and *M. scorpioides*. However, having produced a low total number of flowers (n < 10), the flowering phenology of *E. densa* was not analyzed. The statistical tests performed did not show significant differences in the number of flowers observed on a given day depending on water temperature. The high variability in number of flowers produced among mesocosms (Appendix A) made it unlikely to detect a potential difference, and available results show that there was no difference in the amplitude of flowering due to warmer water temperature for the three species that presented analyzable flowering patterns (Figure 5a).

*L. hexapetala* showed differential flowering patterns depending on water temperature regarding synchrony and duration. When exposed to T0, the flowering period started on day 52, and there was a single flowering peak on day 95. When exposed to T3, the flowering started 13 days sooner and was composed of two successive flowering peaks, a smaller one on day 64 and a second bigger one on day 110 (Figure 5b). The expectation–maximization algorithm estimated the flowering to end at the same time for the two temperature treatments, around day 160. For *M. aquatica*, the flowering patterns were similar between the two temperature treatments. There was a single flowering peak in both temperature conditions, which occurred synchronously between the two treatments, on day 107 of the experiment (Figure 5b). The flowering period lasted for 90 days independent of temperature and was almost over when the experiment was stopped, the model predicting it would have lasted about 10 additional days. For *M. scorpioides*, the expectation–maximization algorithm estimated the flowering period to last for 170 days for plants exposed to T0, starting 10 days before the beginning of the experiment, and lasting about 20 additional days after the end of the experiment. It was composed of two flowering peaks, the first one occurring on day 49, and the second one on day 128 (Figure 5b). When exposed to T3, the flowering period of *M. scorpioides* was 70% shorter and was also composed of two flowering peaks, one on day 49, simultaneous to the first flowering peak at T0, and a second peak on day 79.

## 3. Discussion

In this study, we evaluated growth, some physiological traits related to metabolism, and flowering phenology for six macrophytes species in response to an experimental warming in outdoor mesocosms. The experimental set-up led to an average 3 °C increase in water temperature, as sought for. Depending on species, we observed differences in percent cover, maximum canopy height, flowering time and duration, in anthocyanins, flavonols and total soluble sugars content.

### 3.1. Warmer Water Has Impacts on Growth, Phenology and Metabolism

#### 3.1.1. Few Changes in Morphological Traits Related to Growth

The percent cover of *E. densa* was negatively impacted by the warming treatment. While for that species optimum growth temperature is between 16 and 25 °C [33], in the warming treatment, water temperature reached at least 25 °C for about 20 days during the first half of the experiment that occurred in early summer (Figure 6). A reduction of the dry mass was also expected, but was not observed. The lack of effect on mass at the end of the experiment could be explained by the fact that at lower temperatures such as fall temperatures, warming can have positive impacts on *E. densa* [34]. The high variability in individuals’ growth among mesocosms could also explain this absence of significant difference on plant dry biomass between the two treatments, a result found for all species. As supported by the slight decrease in number of individuals during the experiment, some individuals likely underwent the transplantation stress worse than others, leading to a lower acclimation and implantation, and possibly a weaker development for some, thereby inducing this high variability within the individual final dry biomass. Moreover, the shallow rooting medium may have been limiting in final biomass produced.

However, a significantly greater percent cover and maximum canopy height of *L. hexapetala* were determined. Lambert et al. (2010) [35], as well as Hussner (2010) [36], established that the relative growth rate of *L. hexapetala* was positively correlated with water nutrient content. More recently, Thiébaut et al. (2020) [37] showed that the growth and development of the conspecific *L**udwigia peploides* is maintained at 24 °C, with significant energy allocation to growth and nitrogen uptake. Overall, the optimum development of emerged leaved plants such as *Ludwigia* occurs between 25 and 35 °C, air temperatures that will likely occur during boreal summer in the future. Nonetheless, no significant difference in biomass was observed between the two treatments. High variability of plant dry biomass among mesocosms could explain this result. In addition, *L. hexapetala* may have suffered from nutrient limitation in our experiment if critical plant nutrient contents were altered by the warming treatment. Indeed, the N/P ratio increases with mean temperature in plants [38], along with a decline in N and P leaf in macrophytes [39]. Therefore, potential modifications of plant N and P stoichiometry may have resulted in longer but frailer stems in the warmer conditions, which would explain the absence of biomass increase despite the observed increased percent cover and canopy height. Anyhow, the increased percent cover and maximum canopy height of *L. hexapetala* under warmer temperature could confer it greater competitive abilities, notably through the interception of light.

#### 3.1.2. Warming Effects on Primary and Secondary Metabolisms

Although warmer temperature did not show an impact on the growth of most of the studied species, it influenced their metabolism, with modification in the production of measured metabolites. Lower flavonols contents detected in *L. hexapetala*, *M. aquatica* and *M. scorpioides* correspond to a decrease in the antioxidative capacity from these pigments. The phenylalaline ammonia lyase, the key enzyme in the synthesis of phenolic compounds, is extremely sensitive to environmental modifications such as temperature changes, and its activity can be stimulated both by warming and cooling [40]. Similar to flavonols, content in anthocyanins decreased with water warming in *M. aquatica*. These flavonoids play a protective role against photo-oxidation damages and accumulate at low temperatures in presence of light [41]. Therefore, the cooler fall temperatures experienced by plants exposed to ambient temperature conditions, especially at night, may have induced these greater anthocyanins contents. The tested species were experiencing some stress under the ambient temperature conditions, leading to a stimulation of defense mechanisms and stress tolerance represented by increases in flavonols and anthocyanins contents. However, under the warming treatment, plants were investing their energy into growth, as the greater temperatures were generating less stressful or non-stressful growth conditions, explaining the decrease in the two indices.

As secondary metabolism, primary metabolism was impacted by the 3 °C experimental warming that led to a decrease in total soluble sugars in leaves. This result could be explained by a lower photosynthetate production towards the end of the growing season from accelerated life cycle under warm conditions [42,43]. Considering that CO_2_ solubility decreases as water temperature increases, total soluble sugar levels decrease may be caused by reduced CO_2_ assimilation, especially in submerged species. In addition, exposure to greater temperature might have generated earlier exportation of simple sugars from leaves to other parts of the plants as a strategy to increase carbon storage to survive winter [44]. The high amount of total soluble sugars in *M. scorpioides* possibly corresponds to fructane, a fructose polymer that has the particularity of being soluble, and is known to be accumulated by this species [45]. We hypothesize that fructane was extracted along with soluble simple sugars and that the reaction with hot acid then hydrolyzed fructane in fructose monomers that were dosed.

#### 3.1.3. Impacts of Extended and Shortened Flowering Duration

A longer flowering period, as observed for *L. hexapetala*, increases chances of being synchronized with pollinators’ life cycle, which is important for a species such as *L. hexapetala* that has been reported to be strictly outcrossing [46]. Although this was not observed during the present experiment, a longer flowering duration can also lead to a higher number of flowers produced [47] and altogether an extended flowering period can increase sexual reproductive capacities. In addition, the germination, seedling survival and seedling biomass production of this species have been shown to benefit from warmer temperature [48,49]. Depending on external factors such as pollinator presence, a shorter flowering period due to increased temperature, as observed for *M. scorpioides*, may be detrimental to plant sexual reproductive effort [50]. A potential desynchrony between pollination and flowering period limits the chance of successful pollination, and could also be detrimental for pollinator populations [51]. Furthermore, life history traits and reproductive output can be greatly impacted by the increasing frequency of extreme events during the growing season, such as heat waves or droughts [52]. The overall consequences of changes in flowering phenology due to abiotic factors are key to community structure and ecosystem functioning, but are difficult to predict given the complex interactions with other plant and animal species [53]. The observed flowering phenology patterns observed in mesocosms are consistent with field observation in Europe, with flowering from May to September for *M. scorpioides* [54], and from June to October for *L. hexapetala* [55] in the control conditions. Examination of flowering phenological patterns in the field under warmer conditions would bring additional responses and may confirm or disprove the observed patterns. Investigating other phenological cues would also be of interest to predict population and community responses. Contrary to our first hypothesis, we showed that a 3 °C warming of water in outdoor mesocosms did not influence both the growth, metabolism and phenology on a given tested species.

### 3.2. Comparison between Species Depending on Species Biogeographic Origin and Growth Form

The warming treatment applied led to differences that varied depending on species. Traits measured in all six species did not highlight changes specific to growth form. Indeed, the warming treatment generated a decrease in percent cover for one of the two submerged species, and an increase for one out of the four emergent species. Total final dry biomass was not impacted by the temperature treatment, and this for none of the six species, while a temperature effect was observed for soluble sugars, independent of species. These surprising results could partially be explained by the growth variability among experimental subunits, along with potential allelopathic effects. Although species were not in direct competition, the porosity of the separation between subunits within mesocosms might have exposed individuals to allelopathic compounds from other species [56,57], and the tested species might have thereby influenced each other’s growth, independent of water temperature. The overall growth of *E. densa* may have been negatively impacted by the limited depth of the mesocosms, which could have submitted the species to high light intensities detrimental to its growth [58], although *E. densa* can be observed growing in shallow water conditions [59] and can acclimatize to a variety of light regimes [60]. Flower production and physiological metrics may have been impacted by these conditions as well.

In addition, the variety of measurements made was not adapted to all species depending on growth form, morphological traits or the occurrence of flower production. Thus, not all measures were made on the six studied species, which lowered the possibility of strong comparison between species biogeographic origin and between the two growth forms tested. *L. hexapetala*, *M. aquatica* and *M. scorpioides* were the only species tested for all criteria. Flowering phenology, chlorophyll, anthocyanins and flavonols contents, as well as Nitrogen Balance Index were not measured for submerged species *E. densa* and *P. crispus* nor for the emergent *M. aquaticum*, and maximum canopy height was not measured for the two submerged species. Therefore, it is not possible to provide a global conclusion for our second and third hypotheses regarding the differential impact of water temperature warming depending on species biogeographic origin or growth form. Although mesocosm experiments allow for isolating the impact of a single factor such as temperature, they are a simplification of biotic and abiotic interactions. Although present results cannot directly be extrapolated to predict plant community composition and performance, we can presume that warming of air temperature may have enhanced observed differences, or may have led to other modifications, especially for emergent species.

## 4. Materials and Methods

### 4.1. Plant Collection

We used six macrophyte species to test our hypothesis, three species native to Europe, among which were one submersed and two emergent species—per Schuyler’s classification [61]—(*P. crispus* L., Potamogetonaceae; *M. aquatica* L., Lamiaceae; *M. scorpioides* L., Boraginaceae) and three species native to tropical and subtropical South America and invasive at least in Europe and in North America, with one submerged and two emergent (*E. densa* Planch., Hydrocharitaceae; *L. hexapetala* (Hook and Arn) Zardini, H.Y. Gu and P.H. Raven (syn. *L. grandiflora* subsp. *hexapetala*), Onagraceae, and *M. aquaticum* (Vell.) Verdc, Halogaraceae. All the species were collected in the field in Brittany, northwestern France (Appendix A). The three selected invasive species are among the most threatening freshwater plant invaders, can alter ecosystem properties [59], and their distribution in their invasive ranges is predicted to increase under future environmental conditions [62]. The six tested species can co-occur in the field, particularly in lentic water bodies, and therefore constitute a realistic simplified plant community.

For each species, a hundred clones constituted of a stem with a single apex were collected on the same day in spring (April 2015), likely from the same individual. All fragments had a length comprised between 10 and 20 cm, with a relative homogeneity within each species. Clones of *E. densa*, *L. hexapetala* and *M. aquaticum* were shoots only, while those of *P. crispus* included about 2 cm of rootless rhizome and those of *M. aquatica* and *M. scorpioides* presented roots. Back to the laboratory facility, fragments were randomly assembled in ten lots of ten individuals from the same species and were kept indoors in tap water for 3 days before being transferred in outdoor mesocosms.

### 4.2. Experimental Design

Ten mesocosms (1.50 m length × 1.20 m width × 0.50 m depth) located at the ECOBIO experimental garden facility (University of Rennes 1, France) were separated into six sub-units with frost protection fabric that permit water circulation (Figure 6a and Appendix A). Within each mesocosm, each of the six sub-units was randomly attributed to one of the six study species, so that one mesocosm would contain every species. For sub-units assigned to an emergent species, concrete blocks were placed beneath the frost protection fabric, so that emergent species were exposed to a water depth of 5 cm while adjacent submerged species in the same mesocosm were in 20 cm of water. Within each sub-unit, we placed 2 cm of sand above 1 cm of potting soil (NPK 16-7-15) and installed 10 fragments of the designated species by planting them evenly in sediment. Water was actively circulating with a pumping system set up to maintain a 20 cm water depth in each mesocosm. Plants were acclimated to these conditions for 7 weeks at ambient temperature. During this acclimation period, all individuals were replaced when mortality affected half of the individuals or over (i.e., ≥5 individuals) in a given mesocosm for a given species.

Circulating water was coming to and from two tanks adjacent to the mesocosms. Five randomly chosen mesocosms were dependent on one of the tanks for their water supply, while the five other mesocosms were connected to the other tank. In early June 2015, a large Teflon immersion heater was placed in one of the tanks to increase water temperature, to generate warmer water conditions in five out of ten mesocosms (Figure 6a and Appendix A), with the objective of a +3 °C warming, in accordance with the RCP 6.0 scenario projections for 2100 [7]. Therefore, half of the mesocosms was receiving water at ambient temperature (control—T0) under temperate climatic conditions, and the other half was provided with heated water (experimental warming—T3).

Over the 20 weeks of the experiment duration, there were about 5 weeks where the temperature delta measured between T0 and T3 did not show differences between the two treatments (Figure 6b,c). The first period consisted of warm days where tanks were supplied with extra water to compensate for evaporation, which temporarily leveled water temperature, and was done shortly before the water temperature measurements in mesocosms; in a second period, there was an interruption of water circulation that prevented the experimental warming, due to the failure of multiple pumps that had to be replaced. However, excluding these periods, water temperature during the experiment was on average 3.0 ± 0.7 °C greater in treatment T3 compared to T0.

### 4.3. Repeated Measurements of Growth, Phenology and Water Quality

In each sub-unit, number of individuals (i.e., singly rooted plant fragments), maximum canopy height (for emergent species only), number of open flowers and percent cover by plant species were monitored weekly for 20 weeks, until late October 2015, before senescence and biomass decrease. Water temperature and conductivity were measured in the morning once a week in each mesocosm (YSI Professional Plus, Xylem Inc., Yellow Springs, OH, USA). Water samples were collected in tanks about once a month from April to October to determine pH with a pH probe. Nutrient concentrations were assessed: NO_3_^−^, PO_4_^3−^ (colorimeter tests with reagents HI-93728, HI-93713, photometer HI-83200, Hanna Instrument, Woonsocket, RI, USA) and NH_4_^+^ (spectrophotometry). Liquid fertilizer (NPK 4-6-6) was added three times during the experiment (Appendix A) with the aim to maintain non-limiting nutrients conditions. For the two first inputs, 10 mL were added in each of the ten mesocosms. The third input consisted of 34 mL of liquid fertilizer added in each of the two water tanks. These additions notably aimed to reach 10 mg·L^−1^ of NO_3_^−^ in water.

### 4.4. Estimation of Pigment Content, C/N and Total Biomass

At the end of the experiment, leaf content in chlorophyll, flavonols and anthocyanins, as well as Nitrogen Balance Index (NBI) were measured using a non-destructive method with Dualex Scientific+ device (Force-A, Orsay, France) [63] on three emergent species (*L. hexapetala*, *M. aquatica*, *M. scorpioides*). The Dualex Scientific+ device is a hand-held optical leaf-clip meter, and only the three of the abovementioned species had leaves sufficiently large to perform the measurements, i.e., leaves wider than the sensor. The NBI (from 0 to 100) corresponds to the ratio between chlorophyll and flavonols contents, and rather than a measure of leaf nitrogen content per se, it is more of an indicator of C/N allocation changes due to N deficiency. For these species, in each mesocosm, we performed two measures per individual on two leaves. Then, total biomass from all species was harvested, dried at 65 °C for 48 h, and weighed. The different physiological parameters were selected to complement and explain morphological traits, based on the ability to perform measurements without disturbing plant growth.

### 4.5. Total Soluble Sugar Assays

For each species within each mesocosm, a subsample of dried leaves was ground into a fine powder. To extract solutes, 10 mg of tissue powder were solubilized into 1 mL of absolute ethanol, followed by a boiling water bath until complete ethanol evaporation. The dry extract was resolubilized in 2 mL of deionized water, centrifuged (5 min, 4000× *g*, 4 °C), and the supernatant containing water-soluble solutes was collected. Two hundred microliters of supernatant were mixed with 2 mL of anthrone reagent (75 mg anthrone, 75 mg thiourea, 100 mL 70% sulfuric acid) and incubated for 10 min in a boiling water bath [64]. Each sample was triplicated. Glucose solutions ranging from 0 to 1.8 mM were prepared. After incubation at room temperature for 30 min, samples were agitated, and coloration intensity of samples and glucose solutions was quantified by spectrophotometry (λ = 625 nm) (Jenway UV/Vis spectrophotometer, model 6320D, Staffordshire, UK). Soluble sugar concentrations were estimated using the equation obtained from glucose calibration curve (glucose-equivalent concentrations).

### 4.6. Statistical Analyses

All analyses were performed using R 4.0.0 [65]. We applied linear mixed-effect models on percent cover and maximum canopy height, using package ‘lme4’ [66], with temperature, species and their interaction through time as fixed effects, and mesocosm (or block) as a random effect, i.e., Y ~ (Day) ∗ Species ∗ Temperature + (1|Mesocosm). Model assumptions were checked graphically and with Shapiro–Wilk and Levene’s tests, and showed no need for data transformation. An analysis of deviance was performed on the output of the models with package ‘car’ [67] to test the impact of fixed effects. For each species, the differences between temperature treatment T0 and T3 were established by comparing the overlapping of 95% confidence intervals from the predicted values of the models. In addition, the variability explained by the models was calculated for fixed and random effects using the method developed by Nakagawa and Schielzeth (2013) [68] implemented in the package MuMin [69].

A three-way ANOVA was performed to test the impact of species, water temperature and time (beginning of the experiment—after the acclimation period—vs. end of the experiment) and their interaction on the number of individuals, using package ‘car’ [67]. We performed two-way ANOVAs to evaluate the impact of species and water temperature on individual final dry biomass, chlorophyll, flavonols, anthocyanins, NBI and total soluble sugars. Data homoscedasticity and normality of residuals were checked using Levene’s and Shapiro’s tests, respectively. Individual final dry biomass data were square-root transformed and total soluble sugars data were log-transformed to meet analysis of variance assumptions. Tukey’s HSD test was applied using package ‘agricolae’ [70] for multiple comparisons when *p*-values were significant (≤0.05). 

Student’s *t*-tests were performed to evaluate the impact of temperature on the number of flowers opened at a time t. We plotted kernel density estimates and used package ‘mixtools’ [71] to determine the number and timing of flowering peaks and duration of flowering period with an expectation–maximization algorithm. Plots were generated using package ‘ggplot2’ [72].

## 5. Conclusions

To conclude, 20 weeks of exposure to warmer water temperature generated some changes in the growth, metabolism and phenology of exotic and native emergent and submerged macrophytes, that varied depending on species but not on species biogeographic origin or growth form. Of the six studied species, *L. hexapetala* is the only species whose growth would be favored by warmer water. Physiological measurements indicated that the 3 °C warming modified the antioxidant capacities of some species, and possibly modified nutrient allocation in all six species. Finally, the warming treatment resulted in changes in phenology, with modification of the extent of flowering period for two out of the three species that produced flowers. The present results highlight that climate warming will modify certain traits and characteristics of macrophyte species, with potential impacts on communities, which should be further investigated.

## Figures and Tables

**Figure 1 plants-10-01324-f001:**
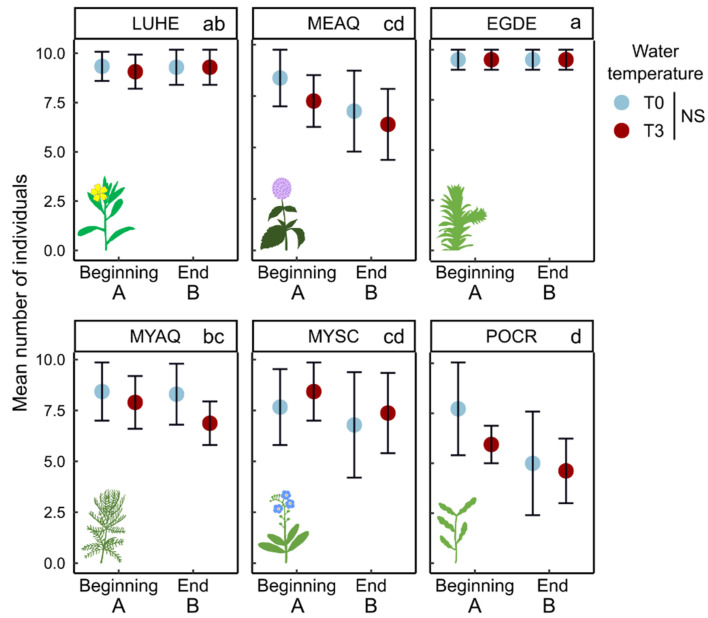
Mean number of individuals (±95% CI) at the beginning and end of the experiment for six species exposed to two temperature treatments (T0 = ambient temperature, T3 = 3 °C experimental warming) over 20 weeks. LUHE = *Ludwigia hexapetala*, MEAQ = *Mentha aquatica*, MYAQ = *Myriophyllum aquaticum*, MYSC = *Myosotis scorpioides*, EGDE = *Egeria densa*, POCR = *Potamogeton crispus*. Small letters indicate differences among species, and capital letters indicate difference between the beginning and the end of the experiment. NS = Not Significant (Three-way ANOVA, n = 5). See detailed credits for plant images in Appendix A.

**Figure 2 plants-10-01324-f002:**
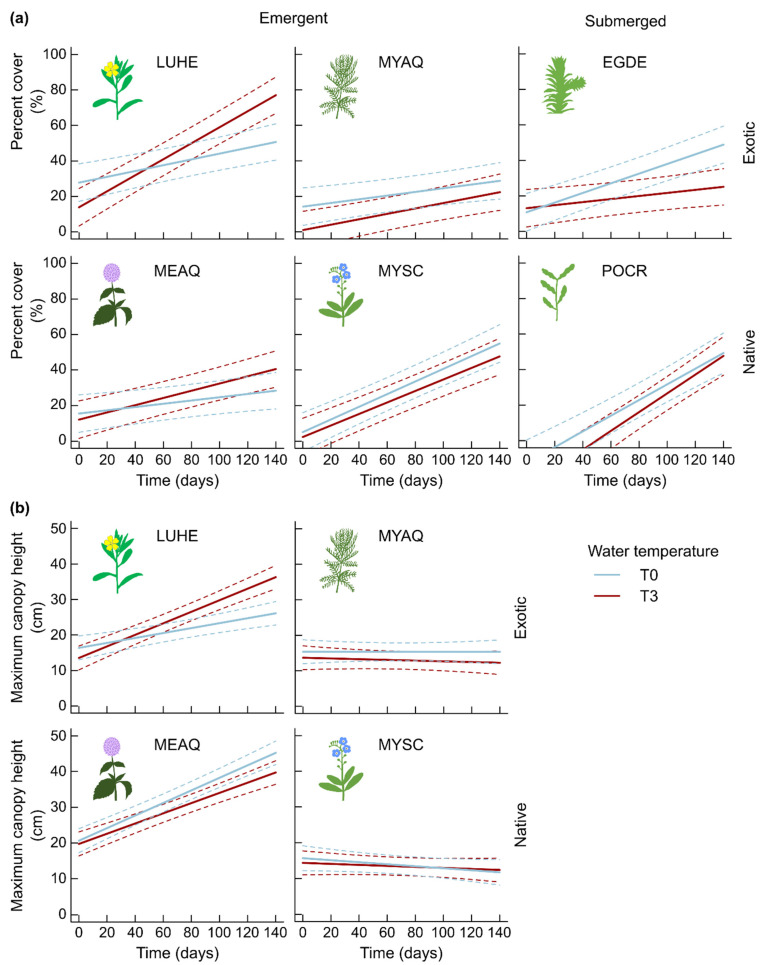
Prediction (±95% CI) of vegetation percent cover by species (**a**) and maximum canopy height of the four emergent species (**b**) over time by linear mixed-effect models for species exposed to two water temperature treatments (ambient temperature T0, 3 °C warming T3) over 20 weeks (n = 5). Conditional R^2^ (fixed and random effects) = 0.61 (**a**) and =0.62 (**b**) The solid lines represent predicted means, and dashed lines represent 95% confidence intervals. LUHE = *Ludwigia hexapetala*, MEAQ = *Mentha aquatica*, MYAQ = *Myriophyllum aquaticum*, MYSC = *Myosotis scorpioides*, EGDE = *Egeria densa*, POCR = *Potamogeton crispus* (POCR).

**Figure 3 plants-10-01324-f003:**
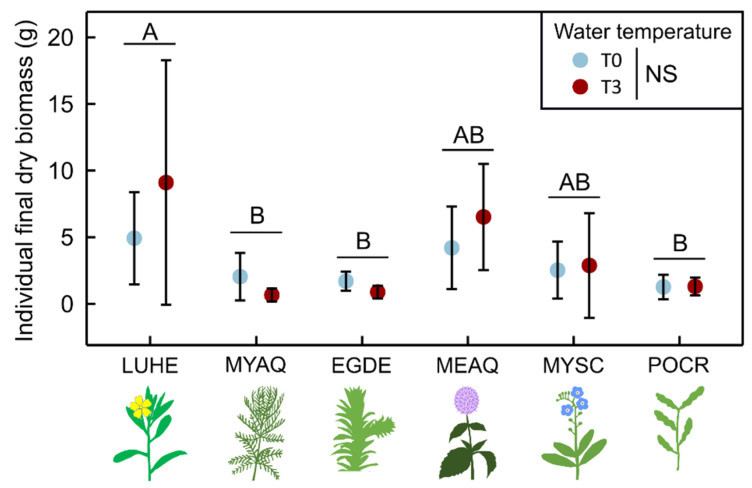
Individual dry biomass (mean ± 95% CI) after 20 weeks of exposure to ambient water temperature (T0) or to a 3 °C warming (T3) for *Ludwigia hexapetala* (LUHE), *Mentha aquatica* (MEAQ), *Myriophyllum aquaticum* (MYAQ), *Myosotis scorpioides* (MYSC), *Egeria densa* (EGDE) and *Potamogeton crispus* (POCR). Capital letters indicate difference among species. NS = Not Significant (Two-way ANOVA, n = 5).

**Figure 4 plants-10-01324-f004:**
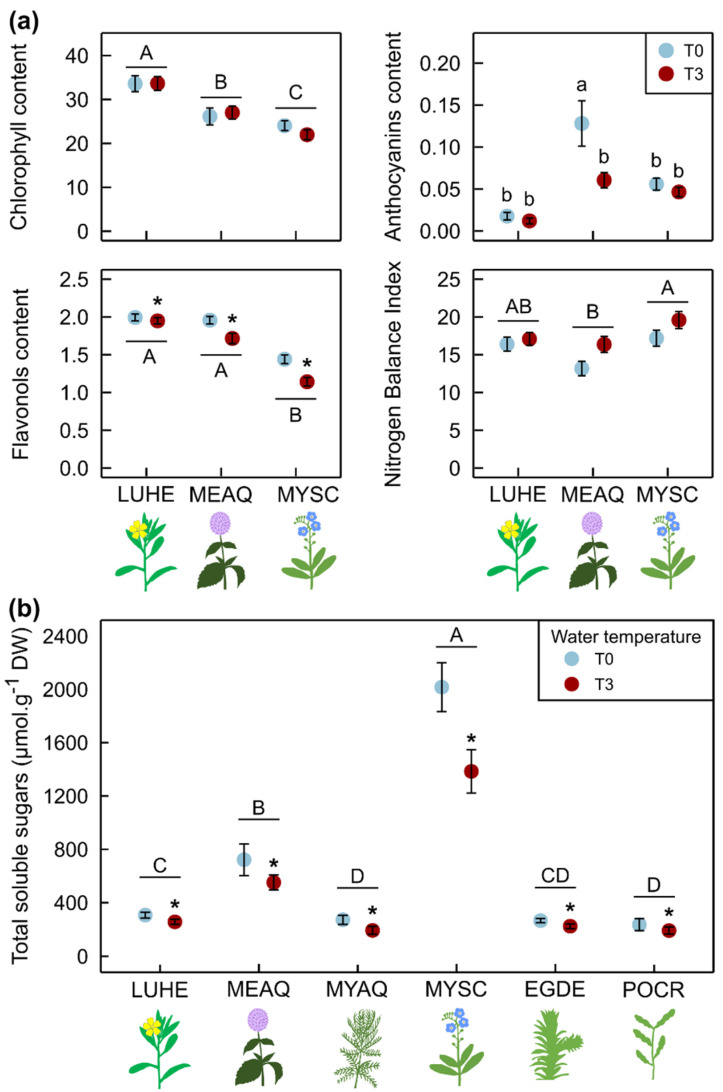
Chlorophyll, anthocyanins, flavonols contents and Nitrogen Balance Index (**a**)—mean ± 95% CI, n = 5) for emergent species with wide enough leaves, and total soluble sugars for six species (**b**)—mean ± 95% CI, n = 15) after exposure to ambient water temperature (T0) and to a 3 °C warming (T3) for 20 weeks. Stars indicate a significant difference compared to control (Two-way ANOVAs). Capital letters show differences among species, and small letters indicate an interaction between species and temperature. LUHE = *Ludwigia hexapetala*, MEAQ = *Mentha aquatica*, MYAQ = *Myriophyllum aquaticum*, MYSC = *Myosotis scorpioides*, EGDE = *Egeria densa*, POCR = *Potamogeton crispus*.

**Figure 5 plants-10-01324-f005:**
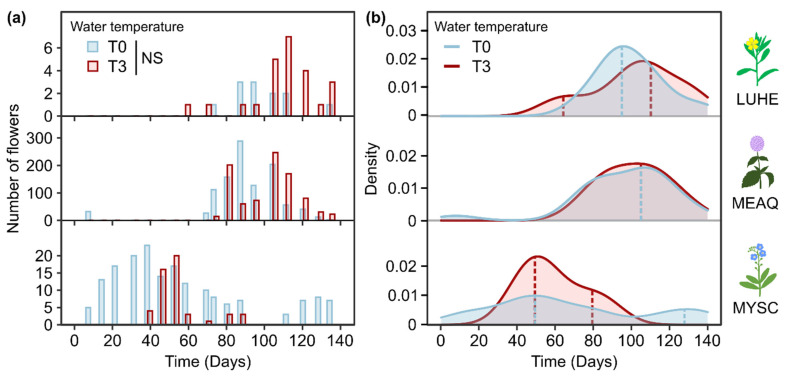
Number of flowers observed over time (**a**) and density of the number of flowers with estimated flowering peaks—dashed vertical lines—(**b**) for *Ludwigia hexapetala* (LUHE), *Mentha aquatica* (MEAQ) and *Myosotis scorpioides* (MYSC) exposed to ambient water temperature (T0) and to a 3 °C warming (T3) for 140 days. NS = Not Significant (multiple *t*-tests, n = 5).

**Figure 6 plants-10-01324-f006:**
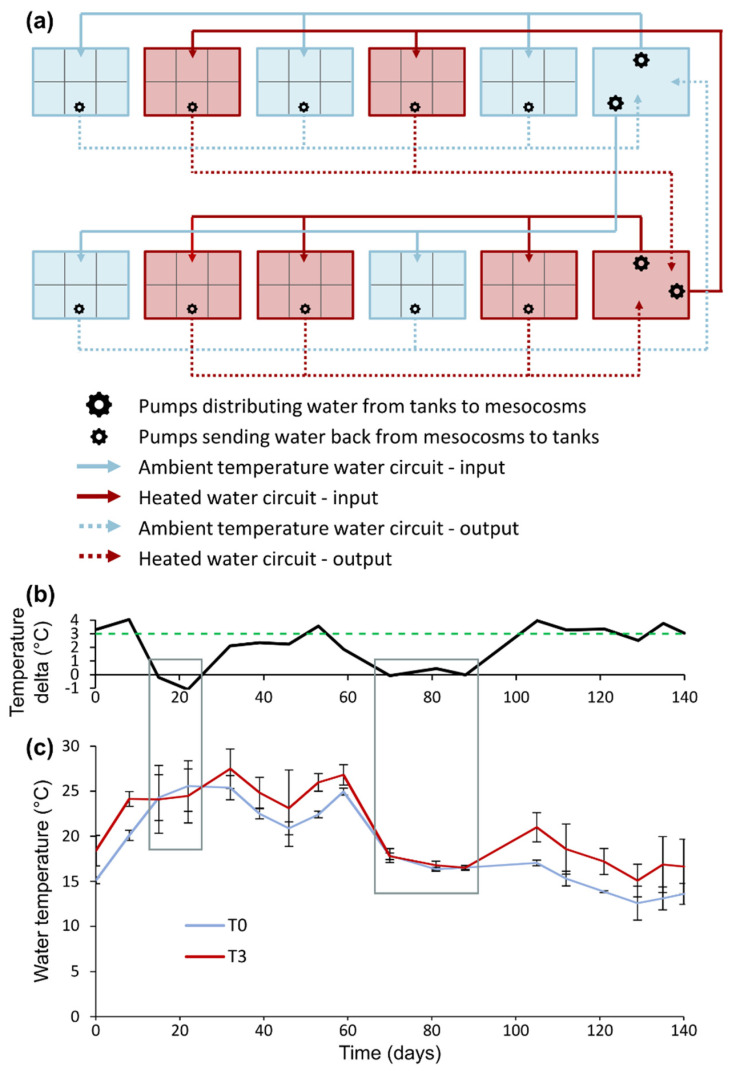
(**a**) Experimental design and water circulation between mesocosms and tanks. (**b**) Mean temperature delta between heated water mesocosms and ambient temperature mesocosms. The green dashed line represents the 3 °C warming objective. (**c**) Mean water temperature (±standard deviation) in heated mesocosms (T3) and ambient temperature mesocosms (T0) during the duration of the experiment. Grey frames highlight two periods when the experimental warming was off due to technical issues.

**Table 1 plants-10-01324-t001:** ANOVA and Tukey HSD test results for the number of individuals of six species exposed to two temperature treatments over 20 weeks.

	% Sum Square	df	*p*
Species	36.3	5	<0.001
Temperature	<0.1	1	0.91
Time	3.8	1	0.01
Species: Temperature	1.2	5	0.83
Species: Time	2.6	5	0.48
Temperature: Time	<0.1	1	0.85
Species: Temperature: Time	0.2	5	0.99
Residuals	55.9	96	

**Table 2 plants-10-01324-t002:** Analysis of deviance results for percent cover and maximum canopy height for six species exposed to two temperature treatments over 20 weeks.

	Percent Cover			Maximum Canopy Height		
	Chi Square	df	*p*	Chi Square	df	*p*
Time	484.51	1	<0.001	78.99	1	<0.001
Species	402.21	5	<0.001	1188.36	5	<0.001
Temperature	0.22	1	0.64	0.38	1	0.54
Time: Species	103.38	5	<0.001	302.93	5	<0.001
Time: Temperature	5.91	1	0.015	4.22	1	0.039
Species: Temperature	71.47	5	<0.001	34.78	5	<0.001
Time: Species: Temperature	47.38	5	<0.001	20.40	5	0.001

## Data Availability

Data supporting reported results will be made available upon acceptance.

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
