# Peer review of "Heterogeneous Impact of Water Warming on Exotic and Native Submerged and Emergent Plants in Outdoor Mesocosms"

_plants, 2021, doi:10.3390/plants10071324_

Round 1

Reviewer 1 Report

plants-1247584

Review Comments:

  1. Ln 80 “Beneficiate”? why not just  “benefit”
  2. Lns 86-89: E. densa: fragment issue; but why not carefully count rooted plants?
  3. 91-92: “number of individuals”- how defined? What is an individual? Singly rooted plants?
  4. Ln 97 and 106: Brasiliense is no longer used: Correct to current nomenclature: M. aquaticum. I realize that creates some confusion with M. aquatica and M. aquaticum, but those are the correct Latin names.
  5. RE all figures and figure captions: please correct to use aquaticum.
  6. Re flowering:

Ln 178-182:  Authors  appear to state the “high variability in mesocosms” caused there to be no differences flowering…ergo there was no difference.  But that’s not correct: The high variability made it unlikely to be able to determine IF there was a difference in flowering.    In other words: the authors could not conclude whether or not there was any significant difference.   There could just as equally been a real difference, but not discernable due to variability.

What is the “model” referred to?  Please cite or explain.

  1. Ln 217-220: How are authors defining “fragments”?   Do they mean actually separate pieces of  shoots (with roots?).  Please define.
  2. Ln 238 and others (e.g. ln 408): please check for multiple periods, or periods in the wrong place.   Suggest a thorough reading of the ms to detect grammatical and typographic errors.
  3. LN 267-273: This is just speculation with no evidence of which sugars accumulated. I suggest delete.
  4. Ln: 294-299: Please rephrase this: Additional field monitoring would provide more data on flowering conditions and help to determine if the mesocosm results reflect changes observed in the field.  They wouldn’t  necessarily  “confirm” results.
  5. Ln 304-307: Please rephrase this more clearly.
  6. LN 307- 316: Again, this is speculation and there is not data from this experimental design to substantiate or negate any allelopathy.
  7. Ln 329- 330: Actually field experiments with invasive plants are done on already established populations.
  8. Ln 330-334: The authors cannot state that just the “rooting system” response led to the observed changes because the entire plant was exposed to the 3 degree rise, or some elevation (in emersed species) between the water and air interface.
  9. Experimental Design:
    The concept of changing a single driver of growth: ambient temperature and determining responses of submersed and emergent aquatic plants is good and could result in valuable data with some relevance to global temperature elevation.
  10. However, I am very concerned with one major problem in this specific design:
    That is the extremely limited depth (20 cm) for examining growth responses in submersed plants that typically grow during one season as much as 1 to >2 m in length, and commonly grow from the sediment to achieve those canopy heights (and attendant biomass). It is not only the limited depth (and therefore limited volume in which the submersed plants can grow) it is also the fact that in the field, light quality changes in the water column with depth, including the red/far red ratios, which influence flowering, canopy structure and vegetative reproductive propagule formation.  
    The limited water depth and resultant light field conditions probably affected several of the physiological metrics measured including pigment content.  It isn’t surprising that the shallow water conditions suppressed E. densa growth and flowering: This plant does not grow well in full sunlight and is highly adapted to low irradiance (50- 200 micro moles/m2  PAR).  What were the light levels during this experiment?  What was the day-length?
  11. It appears that the plant were in a “semi-common garden” conditions: sharing the sediments provided, though partitioned with screens.This has compounded potential plant-plant interactions (as the authors have suggested: potential allelopathy).
  12. The quite shallow sediment (total 3cm if I read the protocol correctly) is very thin and limiting to root and rhizome growth and spread.
  13. The combinations of shallow water, thin planting substrate and their attendant effects makes this study of limited utility- and relevance to “field” conditions for the submersed plants. For the emergent species, the water level is less of concern, but the shallow rooting medium suggests that this may have been limiting in final biomass produced.

I suggest the authors consider limiting their interpretation to the three emergent species and re-write the results and discussion based on that data.

Author Response

Please see the PDF file attached.

Reviewer 2 Report

An interesting and very well written manuscript. I have only very minor comments that I included in the PDF version. I would suggest to publish after minor revision.

Author Response

Please see the PDF file attached.

Reviewer 3 Report

This is a good manuscript that includes a complete analysis of the effect of the temperature on different type of plant traits.

In my opinion, only minor changes are recommended:

  • The first time that the species are mentioned, should be cited with the complete name for a better reading comprenhesion (i.e E. densa in line 86 is the first time that is mentioned, I suggest to use the complete name, Egeria densa).
  • Line 341: correct Egeria densa Planch (author without parenthesis)
  • Figure 1: In the legend it is mentioned that capital letters indicate differences between the beginning and the end of the experiment, but it seems that there are not differences within species, so, is this an error or a misinterpretation?
  • Results: the authors mention that there is a slight decrease for species during the experiment. Could you give some explanation in discussion? Is this a consequence of a low acclimatization?

Author Response

Please see the PDF file attached.

Round 2

Reviewer 1 Report

This paper still lacks critical information if one were to duplicate the conditions (e.g. light levels); and the growing conditions for the two submersed species are not at alll representative of typical/normal habitat conditions for those species.  Perhaps just just include the emergent species.